# Microevolution of *Trypanosoma cruzi* reveals hybridization and clonal mechanisms driving rapid genome diversification

Gabriel Machado Matos[1,2†], Michael D Lewis[3†], Carlos Talavera-López[2,4], Matthew Yeo[3], Edmundo C Grisard[5], Louisa A Messenger[3], Michael A Miles[3], Björn Andersson[2]*

[1]Departamento de Biologia Celular, Embriologia e Genética, Universidade Federal de Santa Catarina, Florianopolis, Brazil; [2]Department of Cell and Molecular Biology, Karolinska Institute, Stockholm, Sweden; [3]Faculty of Infectious and Tropical Diseases, London School of Hygiene and Tropical Medicine, London, United Kingdom; [4]Institute of Computational Biology, Computational Health Centre, Helmholtz Munich, Munich, Germany; [5]Departamento de Microbiologia, Imunologia e Parasitologia, Universidade Federal de Santa Catarina, Florianopolis, Brazil

**\*For correspondence:**
bjorn.andersson@ki.se

[†]These authors contributed equally to this work

**Abstract** Protozoa and fungi are known to have extraordinarily diverse mechanisms of genetic exchange. However, the presence and epidemiological relevance of genetic exchange in *Trypanosoma cruzi*, the agent of Chagas disease, has been controversial and debated for many years. Field studies have identified both predominantly clonal and sexually recombining natural populations. Two of six natural *T. cruzi* lineages (TcV and TcVI) show hybrid mosaicism, using analysis of single-gene locus markers. The formation of hybrid strains in vitro has been achieved and this provides a framework to study the mechanisms and adaptive significance of genetic exchange. Using whole genome sequencing of a set of experimental hybrids strains, we have confirmed that hybrid formation initially results in tetraploid parasites. The hybrid progeny showed novel mutations that were not attributable to either (diploid) parent showing an increase in amino acid changes. In long-term culture, up to 800 generations, there was a variable but gradual erosion of progeny genomes towards triploidy, yet retention of elevated copy number was observed at several core housekeeping loci. Our findings indicate hybrid formation by fusion of diploid *T. cruzi*, followed by sporadic genome erosion, but with substantial potential for adaptive evolution, as has been described as a genetic feature of other organisms, such as some fungi.

## Editor's evaluation

The authors dissected the across-the-genome consequences of sexual recombination in *Trypanosoma cruzi*, a serious human pathogen. They had discovered hybrid formation in this species 20 years ago, here they went at length by culturing parental and hybrid clones for 5 years, demonstrating that tetraploid *T cruzi* hybrids undergo genome erosion.

## Introduction

*Trypanosoma cruzi* is a kinetoplastid protozoan and the etiologic agent of Chagas disease, one of the neglected and highest impact parasitic infections in the Americas (*Pérez-Molina and Molina, 2018*).

Chagas disease is estimated to cause great loss in both health-care costs and disability-adjusted life years (*Lee et al., 2013*). Human migration and specific modes of transmission than the canonical vector-based have led to a spreading beyond its natural geographical boundaries, becoming a global issue (*Pérez-Molina and Molina, 2018*; *Bern, 2015*). *T. cruzi* transmission is a zoonosis maintained by numerous species of triatomine insects and different species of mammals (*Jansen et al., 2018*). This parasite has a complex life cycle, where transmission to humans occurs most frequently by contamination with infected feces from triatomine vectors. Other routes of human infection include congenital, oral (mainly consumption of raw contaminated foods), and contaminated transplant tissues and blood products. To evade the immune responses, the parasite displays a vast, complex repertoire of surface proteins involved in cell invasion and pathogenicity (*De Pablos and Osuna, 2012*; *Pech-Canul et al., 2017*; *Talavera-López et al., 2021*). The regions of the genome coding for these molecules are composed of highly repetitive sequences, with hundreds to thousands of members of each family, as well as substantial numbers of transposable elements, microsatellite, and tandem repeats (*Talavera-López et al., 2021*; *El-Sayed et al., 2005*; *Wang et al., 2021*).

The reproductive mode of these parasitic protozoans has been widely debated, and both preponderate clonal evolution (*Tibayrenc and Ayala, 2012*; *Tibayrenc and Ayala, 2013*) and sexual reproduction (*Ramírez and Llewellyn, 2014*; *Schwabl et al., 2019*; *Inbar et al., 2019*) have been proposed. Many eukaryotic pathogens have essentially clonal population structures while also having non-obligate sexual cycles, which may enable them to adapt to environmental changes (*Heitman, 2006*; *Lewis et al., 2011*; *Steensels et al., 2021*). In trypanosomatids, the most well-studied mating system is that of *Trypanosoma brucei*, for which a meiotic process is well supported based on the identification of a haploid parasite stage in the vector (*Peacock et al., 2014*) and on the patterns of allele inheritance and recombination observed in experimental hybrids (*MacLeod et al., 2005*). In *T. cruzi*, great genetic and phenotypic diversity is observed, and six distinct genetic clades have been recognized, named TcI to TcVI (discrete typing units or DTU-I to -VI) (*Zingales et al., 2012*). Genotyping analysis indicates that TcV and TcVI are recent, natural inter-lineage hybrids of TcII and TcIII, showing that recombination events have shaped the evolution of *T. cruzi* lineages (*Schwabl et al., 2019*; *Lewis et al., 2011*; *Messenger and Miles, 2015*). In addition to natural evidence of hybridization, the formation of *T. cruzi* hybrids in vitro was also described (*Gaunt et al., 2003*). Genetic marker analyses of hybrid strains showed multi-allelic (non-Mendelian) inheritance of microsatellite alleles and an elevated DNA content (*Gaunt et al., 2003*; *Lewis et al., 2009*). The hybridization events in *T. cruzi* were proposed to occur via a parasexual mechanism, similar to those observed in certain fungi (*Lewis et al., 2011*; *Steensels et al., 2021*; *Gaunt et al., 2003*; *Bennett, 2015*). While certain aspects of the molecular mechanisms involved in this process have been studied (*Alves et al., 2018*), the contribution of each parental strain in hybrid genomes, as well as the mechanisms and adaptive significance of the hybridization phenomenon, are not well understood.

Experimental evolution approaches in defined environments have been widely used to study the evolution of model organisms (such as bacteria, yeast, and *Drosophila* sp.) (*Zeyl, 2006*; *Burke and Rose, 2009*; *Kawecki et al., 2012*; *Lenski, 2017*) although this is not the case for parasitic protozoa. To investigate the effects of clonal reproduction and genetic exchange on the *T. cruzi* genome at the microevolutionary scale, we selected two closely related TcI strains (P1, P2) and three hybrid clones that were generated from a P1 × P2 cross (1C2, 1D12, 2C1) (*Gaunt et al., 2003*). In addition, to dissect the underlying mechanisms of hybridization in *T. cruzi,* we applied a comparative genomics approach based on genome sequence data generated for parental strains and hybrid clones at the beginning and at the end of the in vitro microevolution experiment.

## Results
### Stability of DNA content during experimental in vitro microevolution

To investigate the effects of clonal reproduction and genetic exchange on the *T. cruzi* genome at the microevolutionary scale, we selected two TcI strains (P1, P2) and three hybrid clones that were generated from a P1 × P2 cross (1C2, 1D12, 2C1) (*Gaunt et al., 2003*). The two parents and three hybrids were included in an in vitro evolution experiment in which they were continuously cultured as epimastigote forms for 5 years, equivalent to approximately 800 generations of replication by binary fission (*Figure 1A*). At the end of the experiment, we re-cloned the parasite lines and selected three

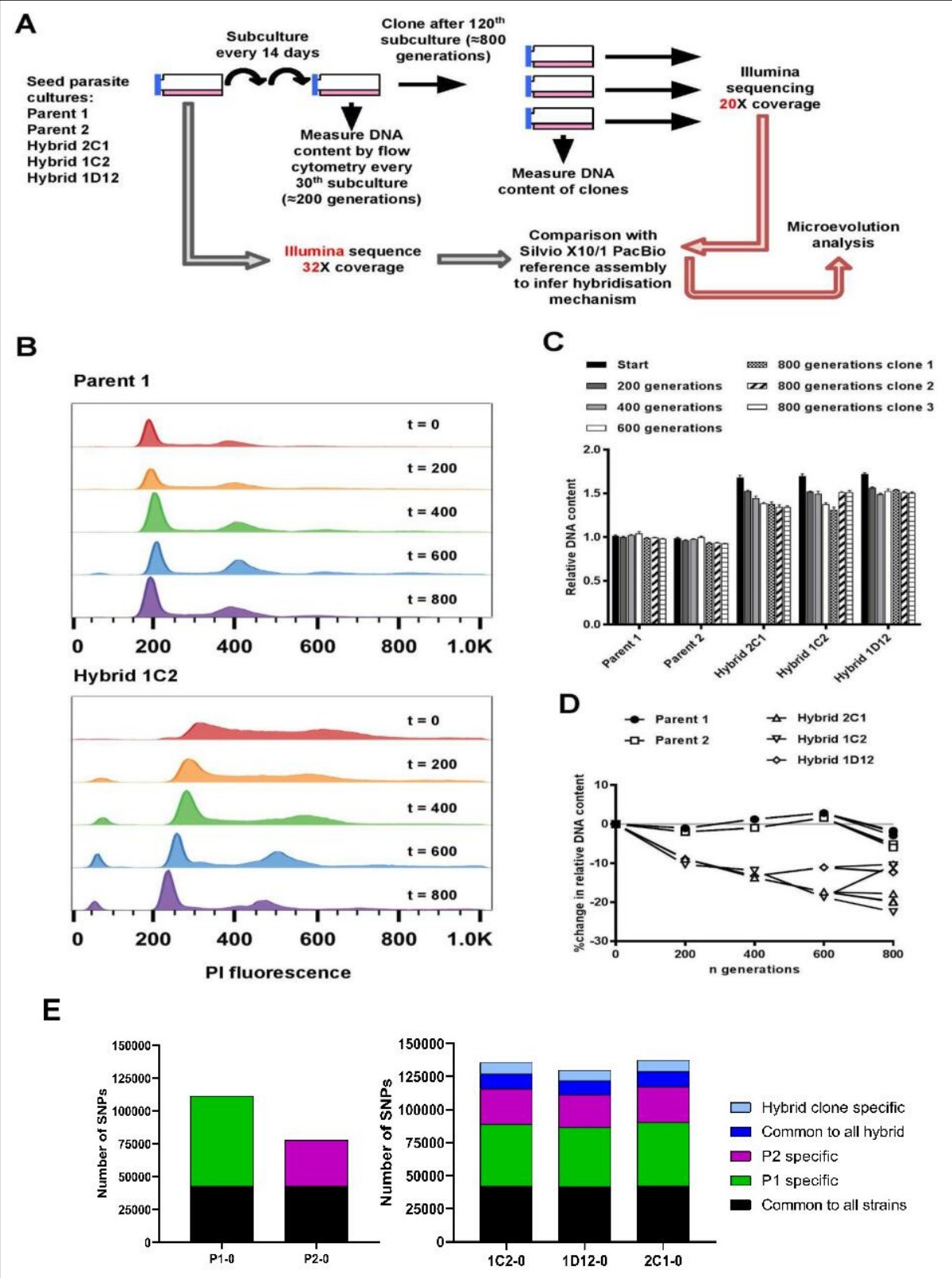

**Figure 1.** In vitro microevolution of *Trypanosoma cruzi* hybrids. (**A**) Overview of experimental design. (**B**) DNA content profiles shown as propidium iodide (PI) staining intensity histograms measured by flow cytometry. PI fluorescence = relative fluorescence units; y-axis shows number of events from the population of parasites at the indicated generation of the in vitro culture experiment from start (t=0) to finish (t=800). Plots of one parent and one hybrid are shown, representative of three biological replicates for each time point. The vertical dashed line marks the position of the G1 fluorescence

*Figure 1 continued on next page*

*Figure 1 continued*

intensity peak at t=0. (**C**) Quantitative analysis of DNA content for both parents and three hybrid strains at each time point and for three clones per strain generated after the final time point of 800 generations. Data are the mean + SEM of three biological replicates after normalization against an internal standard and conversion to a ratio of the mean of the two parents at t=0. The predicted total DNA content ranges equating to tetraploid (4n1k) and triploid (3n1k) parasites are indicated, under the assumption that kDNA constitutes between 20% and 30% of total DNA. (**D**) The same data as shown in C, represented as the % change in DNA content over time compared to t=0. (**E**) Distinct genomic signatures exclusive to each parental strain were inherited by the hybrid strains. In black are represented SNPs present in both parental strains, in green SNPs exclusive to the P1 strain, in purple SNPs exclusive to the P2 strain, in dark blue SNPs common to all hybrids, and in light blue SNPs exclusive to each hybrid clone.

clones per line for analysis. During this microevolution experiment, we monitored the DNA content of each parasite line at approximately 200-generation intervals to check for evidence of large-scale structural genomic changes. We observed that the DNA content of the two parental lines remained approximately constant (*Figure 1B–D*). At the start of the experiment, the total DNA content of the hybrid clones was approximately 70% greater than the parentals (*Figure 1C, D*, *Lewis et al., 2009*), which equated to slightly below expectations for a tetraploid parasite with one kinetoplast (containing mitochondrial DNA), that is, a 4n1k (4C) genome size. Over the course of the microevolution experiment, in contrast to P1 and P2, we found that the DNA content of the hybrids became progressively lower over time, up to a maximum decrease of 22.5% by generation 800. By extrapolation from the estimated genome sizes of P1 (94.5 Mb) and P2 (92.5 Mb) (*Lewis et al., 2009*), these data indicate genome erosion in the hybrids occurred at an average rate of 23 kb per generation. There were, however, notable differences in genome erosion patterns between the three hybrid lines. Hybrid 2C1 showed gradual genome erosion at each time point, with all three 800 generation evolved clones approximating a 3C genome size. Hybrid 1C2 showed a more intermittent pattern – most of the erosion occurred between 0–200 and 400–600 generations, reaching a 3C genome size, but interestingly, 2 of the 800 generation clones gained ~10% more DNA than at t=600. Lastly, hybrid 1D12 had an initial 9% loss between t=0 and t=200, but was then quite stable over time between 3C and 4C.

Next, we looked for evidence of instability at specific loci in the hybrid genomes using PCR-based multilocus microsatellite genotyping, which had previously provided some evidence for non-Mendelian inheritance in the hybrids from the original P1 × P2 cross (*Gaunt et al., 2003*). Specifically, at many loci the hybrids inherited more than one allele per parent. We found that three of the nine evolved hybrid clones had lost at least one microsatellite allele (*Supplementary file 1*). These data indicated that the enlarged genomes of *T. cruzi* hybrids were subject to erosion over an extended period of clonal replication. To dissect the underlying mechanisms, we applied a comparative genomics approach based on sequence data for parental and hybrid samples from the beginning and end of the microevolution experiment.

## Parental strains display distinct genomic signatures

The genomes of the two parental strains were characterized using de novo genome assemblies from short Illumina reads. Despite high sequence coverage and the linking information provided by additional long insert size libraries, the final assemblies only reconstructed 77.1% and 74.8% of the P1 and P2 genomes, respectively (*Supplementary file 2*), due to the high repeat content of the genomes as also observed to other previously sequenced *T. cruzi* genomes. The haploid estimated genome size of P1 differed from P2 by approximately 2 Mbp, and both parental strains genomes were found to be slightly larger than the 44 Mbp reference TcI-Sylvio X10/1 strain (*Talavera-López et al., 2021*), in line with flow cytometry-based DNA content measurements (*Lewis et al., 2009*).

While the parental strains both belong to the TcI clade, with highly conserved synteny in the core regions, they still show significant genetic diversity. Reads from the parental strains were mapped to the reference TcI-Sylvio X10/1 genome, followed by SNP calling and a

**Table 1.** Average SNP density per kilobase in starting generation parental strains and hybrid clones.

|  | SNP/kb genome-wide | SNP/kb high-quality mapping regions |
|---|---|---|
| P1-0 | 9.28 | 2.84 |
| P2-0 | 3.71 | 2.11 |
| 1C2-0 | 7.63 | 3.48 |
| 1D12-0 | 7.46 | 3.29 |
| 2C1-0 | 8.22 | 3.49 |

comparative analysis. The resultant genome-wide average SNP density was 9.28 and 3.71 SNP/kb for P1 and P2, respectively (**Table 1**). Most SNPs were located within repetitive regions, particularly in areas containing surface molecule gene family members, which can be related to gene expansion in these regions. Indeed, copy number variation (CNV) analysis showed that P1 displays these tandem-repeated and surface molecule-coding regions expanded in comparison to P2. To avoid the influence of low mapping quality in repeated regions, we applied a strict mapping quality filter, removing any SNP in regions with mapping quality below 50. SNP density in the retained high-quality regions was 2.84 and 2.11 SNPs/kb for P1 and P2, respectively (**Table 1**), and a total of 68,616 and 34,843 SNPs exclusive for each parental strain were catalogued for use in the analysis of the hybrid strains (**Figure 1E**). This level of differentiation between P1 and P2 provided ample scope to explore the genetic composition and inheritance patterns in the hybrid clones.

Mitochondrial DNA (kDNA) comprises up to 20–30% of the total *T. cruzi* DNA, consisting of relatively conserved dozens of maxi- and thousands of minicircles (**Souza et al., 2011**). This range was used to infer 3C and 4C genome sizes (**Figure 1E**) and the frequency of kDNA-mapping reads was 10.1% for parents and 11.5% for hybrids in average. However, there was substantial variation between samples as expected, and no correlation with the flow cytometric measurement of total DNA content. Despite the large amounts of kDNA maxi- and minicircles/cell, our interpretation is that extraction efficiency of kDNA is lower than for nuclear DNA and more variable between samples, probably due to the highly intercalated molecular network structure of kDNA. All kDNA-mapping reads were removed prior genetic diversity analysis.

## An increase in genetic diversity is observed after hybridization

To evaluate the impact of hybridization on genetic diversity, we compared variant densities in the high-quality mapping regions of the genomes of all hybrid and parental strains. The average SNP density in these regions was higher in all hybrids than in parental strains: 3.48 SNP/kb for 1C2-0, 3.29 SNP/kb for 1D12-0, and 3.49 SNP/kb for 2C1-0 (**Table 1**), partially due to the presence of alleles from both parental strains. In addition to the SNPs clearly inherited from each parental strain, additional SNPs common to all hybrids and SNPs exclusive to each hybrid clone were identified in both coding and non-coding regions, revealing sample-specific patterns (**Figure 1E**). Some SNPs that were common to all hybrid strains were not detected in the parental genomes, mainly consisting in mutations that occurred in culture during the approximately 70 generations of parent parasite growth before the cross was set up (**Figure 1A**). They are unlikely to be attributable to growth of a common progenitor hybrid prior to the cloning step because of differential inheritance of kDNA in 1D12 vs. 2C1 and 1C2 (**Gaunt et al., 2003**), which suggests they are derived from at least two independent hybridization events. Interestingly, de novo mutations common to all hybrids and exclusive to each hybrid clone were predicted to generate a higher percentage of non-synonymous changes in comparison to the mutations inherited from the parental strains (**Supplementary file 3**). Indeed, the ratio of non-synonymous changes per synonymous changes was higher in the de novo mutations common to all hybrids (2.36 for 1C2-0, 2.25 for 1D12-0, and 2.19 for 2C1-0), and exclusive to each hybrid clone (2.40 for 1C2-0, 2.23 for 1D12-0, and 2.37 for 2C1-0) than in those inherited from the parental strains (1.41 for 1C2-0, 1.40 for 1D12-0, and 1.42 for 2C1-0) (**Supplementary file 3**).

To investigate the impact of these mutations on the variable regions of the genome, we compared the ratio of surface protein-coding genes per non-surface protein-coding genes. This ratio is approximately 4-fold higher in SNPs common to all hybrids and in SNPs exclusive to each hybrid clone than in those inherited from the parental strains (**Supplementary file 4**). In addition, we compared the ratio of non-synonymous mutations in surface protein-coding genes per non-synonymous mutations in other genes. Interestingly, this ratio was approximately 3.3-fold higher in the hybrid clone-specific SNPs in comparison to those inherited from the parental strains (**Supplementary file 5**).

## CCNV analysis reveals tetraploid hybrids

Chromosome copy number variation (CCNV) was determined using the combination of read depth coverage (RDC) and allele balance (AB) methods (**Reis-Cunha and Bartholomeu, 2019**). An increase or decrease in the mean RDC of a chromosome when compared to the overall genome coverage is an indicator of a gain or loss of chromosomal sequences, respectively (**Reis-Cunha and Bartholomeu, 2019**). If the ratio between the median chromosome coverage and the median genome coverage

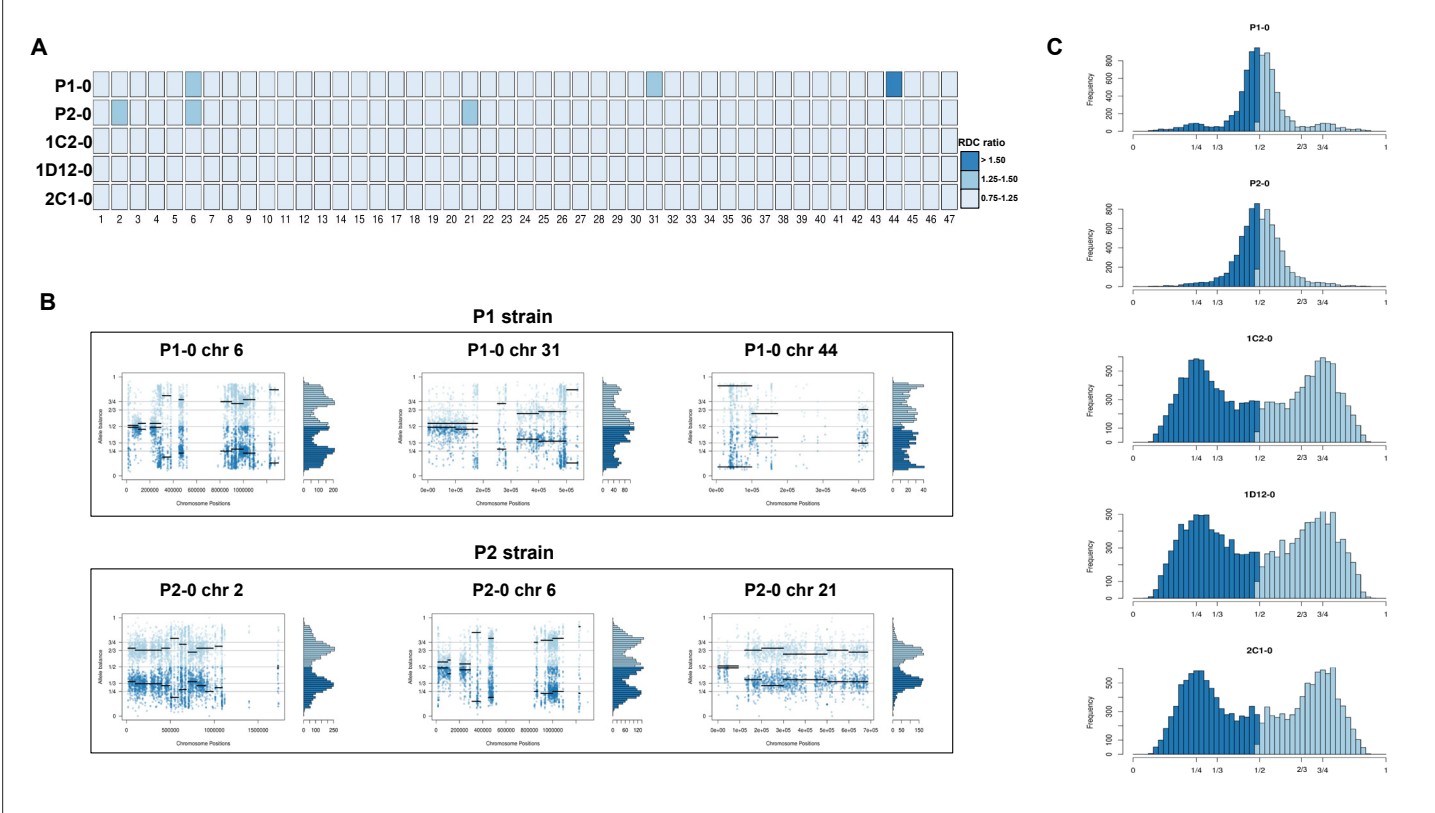

**Figure 2.** Somy estimation based on read depth coverage (RDC) and allelic balance (AB) reveals aneuploidies in parental strains and tetrasomic hybrid clones. (**A**) Aneuploidy analysis based on RDC among all 47 chromosomes in the starting generation of parental strains (P1-0 and P2-0) and hybrids (1C2-0, 1D12-0, and 2C1-0). An increase or decrease in the mean RDC of a chromosome when compared to the genome coverage is an indicator of a gain or loss of chromosomal sequences. When the ratio between the median chromosome coverage and the median genome coverage was approximately one, the chromosome was considered having the same somy as the genome, while fluctuations in this ratio (lower than 0.75 or higher than 1.25) were considered as putative aneuploidies. (**B**) Proportion of the alleles in heterozygous SNP positions in chromosomes with RDC ratio higher than 1.25 to confirm somy estimations. Blue points represent the proportion of the alleles in heterozygous SNP positions along the chromosome. Darker blue represents the frequency of the first allele while lighter blue represents the frequency of the second allele, black lines represent the median value in windows. (**C**) Ploidy estimation based on the proportion of the alleles in heterozygous SNP positions of single-copy genes. It is expected that diploid genomes display a proportion of each allele around 50%, triploid genomes around 33.3% and 66.6%, while tetraploid genomes around 25% and 75% or 50% (**Lenski, 2017**). Darker blue represents the frequency of the first allele while lighter blue represents the frequency of the second allele.

is approximately one, the chromosome has the same copy number as the genome, while fluctuations in this ratio indicate aneuploidies. Here, the RDC analysis was based on the ratio between the mean coverage of predicted single-copy genes in each chromosome and the mean coverage of all single-copy genes in the genome as described by **Reis-Cunha et al., 2015** (**Figure 2A**). In this CCNV estimation, values of '0.5', '1', and '2' denote that the chromosome has, respectively, '0.5', '1', or '2' copies per haploid genome. This means that, if the studied genome is mainly diploid, a value of '1' in this estimation represents two chromosomal copies (one per haploid genome), whereas a value of '1.5' represents three copies and a value of '2' represents four copies. This methodology was shown to eliminate bias caused by chromosomal repetitive content in *T. cruzi* (**Reis-Cunha et al., 2015**), which would otherwise obscure the correct copy numbers. We were unable to estimate CCNV in chromosomes 17, 22, 30, and 47 by this methodology due to the lack of single-copy genes in these chromosomes, and therefore the chromosomal somy prediction presented for them was estimated based on the ratio between the mean RDC of each chromosome position and the mean coverage of all genome positions (**Figure 2A**). Aneuploidy was observed in the P2 strain, involving trisomy of chromosomes 2 and 21 (**Figure 2A, B**). The RDC analysis indicated aneuploidy in chromosome 44 in the P1 strain, but due to the lack of regions with high-quality SNPs, it was not possible to confirm this in the AB analysis (**Figure 2A, B**). The RDC analysis also indicated aneuploidies in chromosome 6 in

both parental strains and in chromosome 31 of P1 strain (*Figure 2A*). However, the AB analysis was not consistent across the entire chromosome (*Figure 2B*), which may be related to CNV in specific regions of these chromosomes rather than aneuploidy, as observed in other *T. cruzi* strains (*Schwabl et al., 2019*; *Reis-Cunha et al., 2015*).

RDC analysis does not allow differentiation between different levels of euploidy (or near-euploidy), therefore whole genome ploidy was estimated by analysing the proportion of the alleles in heterozygous SNP positions of all single-copy genes (*Figure 2C*). In both parental strains, a peak at 50% was observed, while in the hybrids, peaks at 25% and 75% or 50% were observed (*Figure 2C*), which is expected for diploid and tetraploid genomes, respectively (*Reis-Cunha and Bartholomeu, 2019*). This ploidy estimation is in accordance with the previous DNA content estimation based on flow cytometry data, in which the hybrids displayed ~70% greater DNA content than the parental strains (*Figure 1C*, *Lewis et al., 2009*), given that this method could not determine the relative amounts of mitochondrial kDNA and nuclear DNA. Inheritance of SNPs exclusive to both parental strains were observed in the hybrid genomes (*Figure 1E*), which shows that the tetrasomic profile observed resulted from a contribution from both parental chromosomes.

In order to address chromosome specificities and confirm the somy estimates, the AB analysis was performed independently for each chromosome. The proportion of each allele with heterozygous SNPs per chromosome position was plotted for 46 TcI chromosomes (*Figure 3—figure supplements 1–5*), thus excluding chromosome 17 due to the absence of coding regions. Clear patterns of somy were observed in 37 chromosomes (chromosomes 1–16, 18–19, 21, 23, 25–29, 31–33, 35–39, 41, 43, 45–46; *Figure 3—figure supplements 1–5*). Due to a lack of regions containing high-quality heterozygous SNPs generating inconsistent AB, somy estimation was not reliable for chromosomes 17, 20, 22, 24, 30, 34, 40, 42, 44 (*Figure 3—figure supplements 1–5*). Based on our filtering criteria, all SNPs called in chromosome 47 were removed from further analysis, therefore no AB plot is presented for this chromosome. AB analysis shown to be reliable for estimating aneuploidy in chromosomes with longer core regions that contain housekeeping genes and other genes that are conserved between kinetoplastids, while the presence of large tandemly repeated regions and surface protein-coding gene families made read mapping, and thus copy number estimation difficult for some chromosomes. High resolution representations of the CCNV analysis are shown for six chromosomes (*Figure 3*). Chromosomes 2, 7, and 13 have longer tandem repeated regions and coding sequences for surface proteins (percentage of surface protein-coding genes: chr2 7.26%, chr7 9.55%, chr13 8.38%), while chromosomes 18, 21, and 37 have longer core regions (percentage of surface protein-coding genes: chr18 1.25%, chr21 1.98%, chr37 3.74%) (*Figure 3A*). The mapping quality in surface protein-coding sequences was often observed to drop significantly, which affected the inferred proportions of alleles in those regions, and this was found to lead to erroneous somy estimates (*Figure 3B*). To avoid this, a strict mapping quality filter was applied, and all the allele proportions were plotted by chromosome position. Using this strategy, it was possible to identify specific regions in chromosomes that could be used to infer CCNV with high confidence. Chromosomes 2, 7, and 13 showed longer regions with diverse allele proportions, nonetheless it was still possible to identify the trisomy of chromosome 2 of P2 and tetrasomies in hybrids based on using the smaller core regions (*Figure 3C*). Chromosomes 18, 21, and 37 showed somy patterns that were clearly consistent with the whole genome CCNV, confirming the trisomy in chromosome 21 of P2 and tetrasomy in the hybrid clones (*Figure 3D*).

## CCNV reveals sequential loss of chromosomal copy in hybrid clones after culture growth

CCNV and whole genome ploidy were evaluated in three replicate clones for each parental and hybrid strains after 800 generations in continuous in vitro culture. The trisomies in chromosomes 2 and 21 of P2 (*Figure 2A*) were no longer observed in any replicate clone after growth in culture (*Figure 4A*). New aneuploidies were identified in the parental strains, involving trisomy of chromosomes 37 and 45 in all P1 clones and trisomy of chromosomes 32 and 37 in all P2 clones (*Figure 4A, B*). As observed in the first generation, the RDC and AB analyses in chromosomes 6 and 31 were not consistent throughout the chromosomes (*Figure 5—figure supplements 1–15*). Whole genome ploidy estimation revealed a shrinking pattern in the hybrids, which was consistent with the prior flow cytometry analysis (*Figure 1C*). While parental strains remained essentially diploid (apart from the few aforementioned trisomies), a mixture between trisomic and tetrasomic profiles was observed in

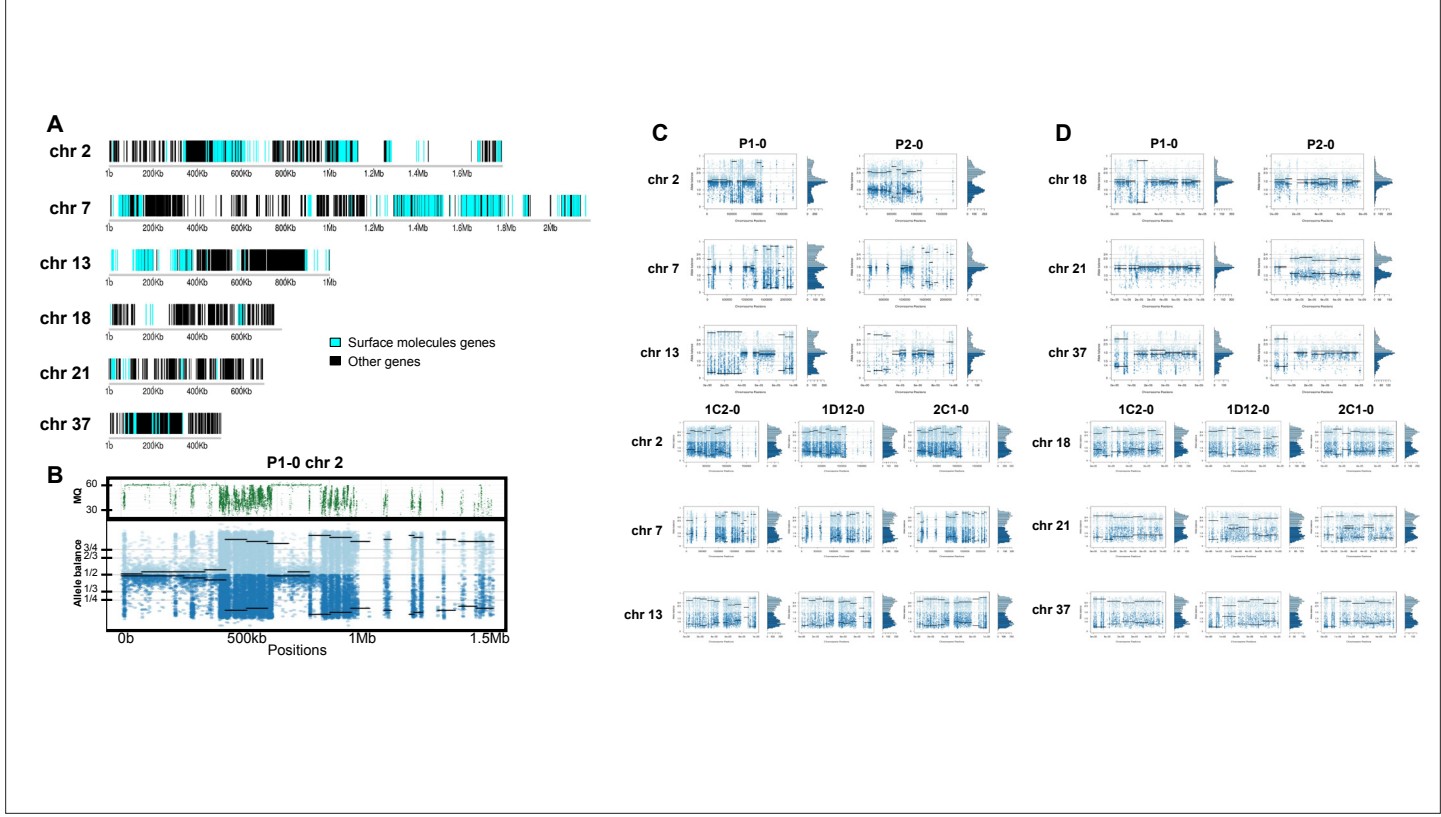

**Figure 3.** Somy estimation per chromosome based on allele balance (AB) in starting generation confirming aneuploidies in parental strains and tetrasomic hybrid clones. (**A**) Gene distribution and its implication in mapping quality (MQ) and somy estimation of six representative TcI chromosomes with longer surface molecule-encoding regions (chromosomes 2, 7, and 13) and shorter encoding surface molecule-encoding regions (chromosomes 18, 21, and 37). Surface protein-coding genes are represented as cyan blue boxes while other genes are represented as black boxes. (**B**) Implication of chromosome gene distribution in MQ and somy estimation based on AB in chromosome 2 (longer surface molecule-encoding regions) of parental strain 1 (P1-0). Dark green represents the fluctuation in MQ along chromosome positions. Blue points represent the proportion of the alleles in heterozygous SNP positions along the chromosome. Darker blue represents the frequency of the first allele while lighter blue represents the frequency of the second allele, black lines represent the median value in windows. (**C**) Proportion of the alleles in heterozygous SNP positions in chromosomes with longer surface molecule-encoding regions. (**D**) Proportion of the alleles in heterozygous SNP positions in chromosomes with shorter surface molecule-encoding regions. Blue points represent the proportion of the alleles in heterozygous SNP positions along the chromosome. Darker blue represents the frequency of the first allele while lighter blue represents the frequency of the second allele, black lines represent the median value in windows.

The online version of this article includes the following figure supplement(s) for figure 3:

**Figure supplement 1.** Somy estimation per chromosome based on allele balance (AB) in starting generation confirming aneuploidies in essentially diploid P1.

**Figure supplement 2.** Somy estimation per chromosome based on allele balance (AB) in starting generation confirming aneuploidies in essentially diploid P2.

**Figure supplement 3.** Somy estimation per chromosome based on allele balance (AB) in starting generation confirming tetrasomic chromosomes in hybrid 1C2.

**Figure supplement 4.** Somy estimation per chromosome based on allele balance (AB) in starting generation confirming tetrasomic chromosomes in hybrid 1D12.

**Figure supplement 5.** Somy estimation per chromosome based on allele balance (AB) in starting generation confirming tetrasomic chromosomes in hybrid 2C1.

the hybrids after growth in culture (*Figure 4C*). Indeed, hybrid 2C1 displayed a clear transition from a tetraploid to a triploid pattern after growth in culture (*Figure 4C*), suggesting that genome erosion eventually leads to a return towards diploidy after the hybridization event, as observed in the naturally occurring TcV and TcVI hybrids (*Lewis et al., 2009*). In contrast with the first generation, the RDC analysis indicates putative aneuploidies in hybrid clones after extended culture growth (*Figure 4A*). Despite the patterns of genome erosion, tetrasomic profiles in chromosomes 19 and 37 were present

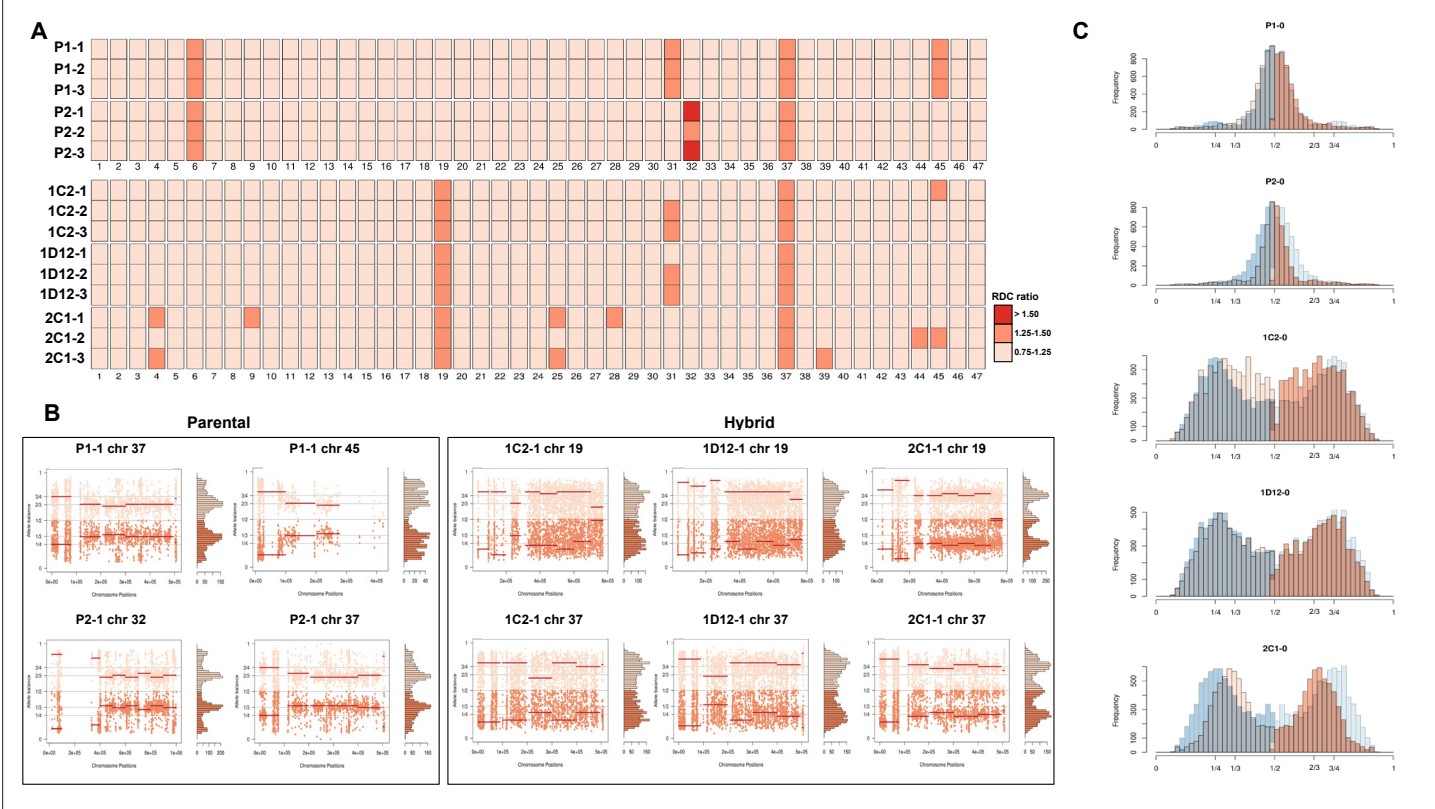

**Figure 4.** Somy estimation based on read depth coverage (RDC) and allele balance (AB) reveals novel aneuploidies and a shift in hybrid strains ploidy after 800 generations in in vitro culture. (**A**) Aneuploidy analysis based on RDC among the 47 chromosomes in all clones after culture growth. (**B**) Proportion of the alleles in heterozygous SNP positions in chromosomes showing aneuploidies in all parental or hybrid clones. (**C**) Comparison of somy estimation based on the proportion of the alleles in heterozygous SNP positions of single-copy genes. First generation is presented in shaded blue, while the 800 generation is presented in orange.

in all hybrid clones after culture growth (*Figure 4A, B*). Taken together with previous DNA content analysis of intermediate time points (*Figure 1*), our results from the different in vitro evolved clones indicate that the genome erosion seems to be gradual phenomena, with sequential losses of chromosome copies and regions rather than coordinated jumps between ploidy levels. The proportion of each allele in heterozygous SNPs per chromosome position was also plotted for the 47 chromosomes in each replicate after culture growth (*Figure 5—figure supplements 1–15*). As observed in the representative chromosomes, the evolved parental clones still have a clear disomic pattern, while a mixture of trisomic and tetrasomic patterns is observed in evolved hybrid clones (*Figure 5*; *Figure 5—figure supplements 1–15*).

## Patterns of genome erosion after in vitro microevolution

To investigate the patterns of genome erosion and CCNV in the hybrids, we performed a CNV analysis throughout the whole genome of parental and hybrid clones after extended culture growth. Despite the stability of DNA content in parental clones during our experiment (*Figure 1C, D*), similar gene categories showed CNV in both parental and hybrid clones after culture growth (*Supplementary files 6–12*). Gene losses occurred mainly in host-parasite interaction genes such as surface protein-coding genes and genes related to exocytosis and cell secretion, while genes related to macromolecule metabolism, ion transport, and cell division showed an increase in copy numbers (*Figure 6*; *Figure 6—figure supplement 1*). Despite the events of genome erosion, chromosomes 19 and 37 remained tetrasomic in all hybrid clones and an extra copy of chromosome 37 is observed in all parental clones after culture growth (*Figure 4B* and *Figure 5B*). Chromosomes 19 and 37 contain long core regions with abundant housekeeping genes involved in cellular metabolism, DNA replication and transcription (*Supplementary files 10 and 11*) which may contribute to fitness in culture. Altogether,

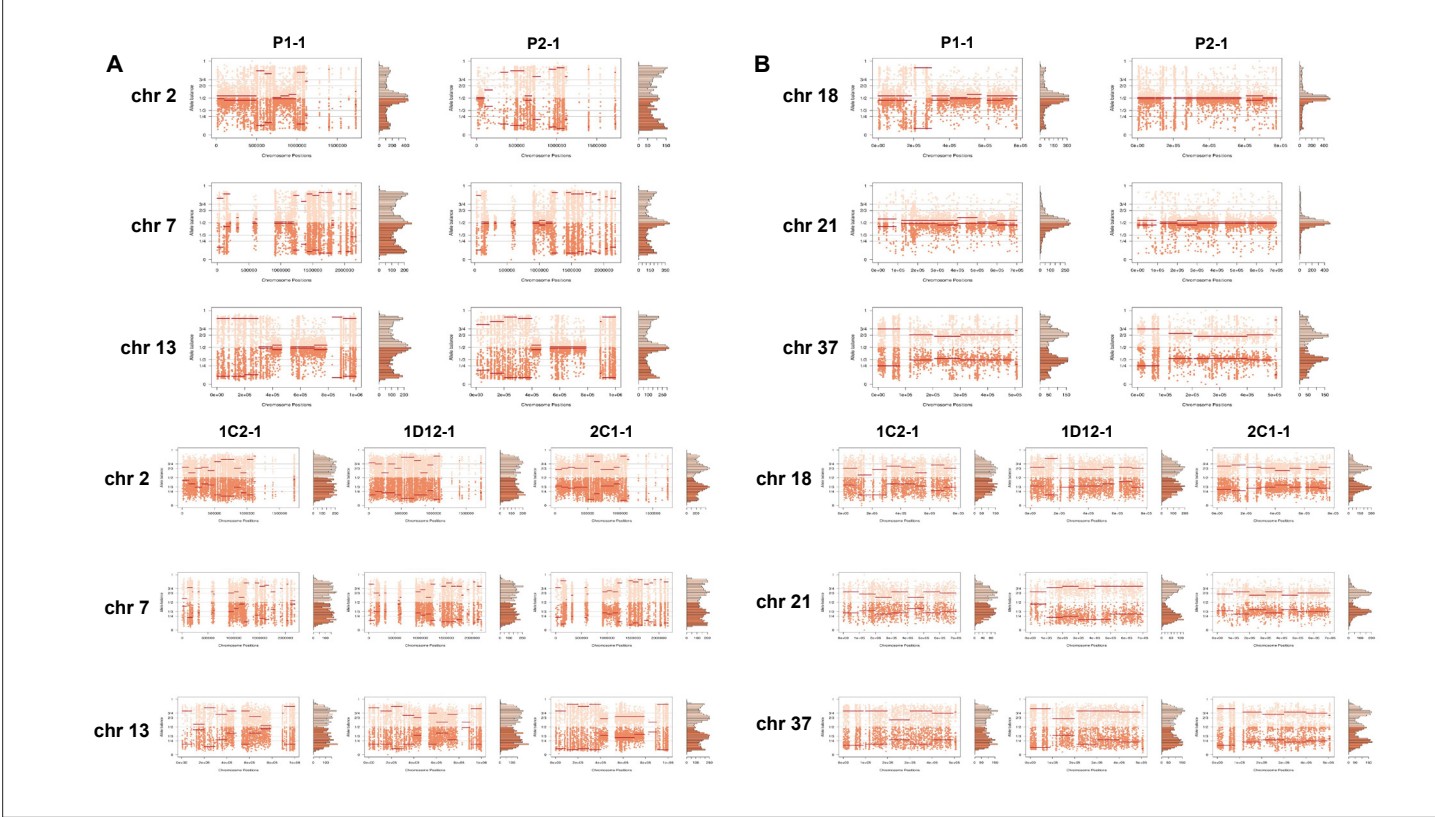

**Figure 5.** Somy estimation confirming novel aneuploidies after in vitro culture. (**A**) Proportion of the alleles in heterozygous SNP positions in chromosomes with longer surface molecule-encoding regions. (**B**) Proportion of the alleles in heterozygous SNP positions in chromosomes with shorter surface molecule-encoding regions. Points represent the proportion of the alleles in heterozygous SNP positions along the chromosome.

The online version of this article includes the following figure supplement(s) for figure 5:

**Figure supplement 1.** Somy estimation per chromosome based on allele balance (AB) confirming novel aneuploidies after in vitro culture in the essentially diploid replicate P1-1.

**Figure supplement 2.** Somy estimation per chromosome based on allele balance (AB) confirming novel aneuploidies after in vitro culture in the essentially diploid replicate P1-2.

**Figure supplement 3.** Somy estimation per chromosome based on allele balance (AB) confirming novel aneuploidies after in vitro culture in the essentially diploid replicate P1-3.

**Figure supplement 4.** Somy estimation per chromosome based on allele balance (AB) confirming novel aneuploidies after in vitro culture in the essentially diploid replicate P2-1.

**Figure supplement 5.** Somy estimation per chromosome based on allele balance (AB) confirming novel aneuploidies after in vitro culture in the essentially diploid replicate P2-2.

**Figure supplement 6.** Somy estimation per chromosome based on allele balance (AB) confirming novel aneuploidies after in vitro culture in the essentially diploid replicate P2-3.

**Figure supplement 7.** Somy estimation per chromosome based on allele balance (AB) confirming trisomic and tetrasomic chromosomes after in vitro culture in replicate 1C2-1.

**Figure supplement 8.** Somy estimation per chromosome based on allele balance (AB) confirming trisomic and tetrasomic chromosomes after in vitro culture in replicate 1C2-2.

**Figure supplement 9.** Somy estimation per chromosome based on allele balance (AB) confirming trisomic and tetrasomic chromosomes after in vitro culture in replicate 1C2-3.

**Figure supplement 10.** Somy estimation per chromosome based on allele balance (AB) confirming trisomic and tetrasomic chromosomes after in vitro culture in replicate 1D12-1.

**Figure supplement 11.** Somy estimation per chromosome based on allele balance (AB) confirming trisomic and tetrasomic chromosomes after in vitro culture in replicate 1D12-2.

*Figure 5 continued on next page*

*Figure 5 continued*

**Figure supplement 12.** Somy estimation per chromosome based on allele balance (AB) confirming trisomic and tetrasomic chromosomes after in vitro culture in replicate 1D12-3.

**Figure supplement 13.** Somy estimation per chromosome based on allele balance (AB) confirming trisomic and tetrasomic chromosomes after in vitro culture in replicate 2C1-1.

**Figure supplement 14.** Somy estimation per chromosome based on allele balance (AB) confirming trisomic and tetrasomic chromosomes after in vitro culture in replicate 2C1-2.

**Figure supplement 15.** Somy estimation per chromosome based on allele balance (AB) confirming trisomic and tetrasomic chromosomes after in vitro culture in replicate 2C1-3.

these results suggest that the presence or absence of selective pressure from the environment may shape CCNV and genome erosion in *T. cruzi*, resulting in distinct genome patterns after hybridization events occur.

## Genetic diversity in evolved genomes

The effect of extended culture growth on genetic diversity was evaluated in all parental and hybrid evolved clones. Regarding the parental strains, all P1 replicates showed a similar SNP density in the retained high-quality regions after culture growth (P1-0: 2.84; P1-1: 2.80; P1-2: 2.77; P1-3: 2.82), while all P2 replicates showed an increase in SNP density after culture growth (P2-0: 2.11; P2-1 2.78; P2-2

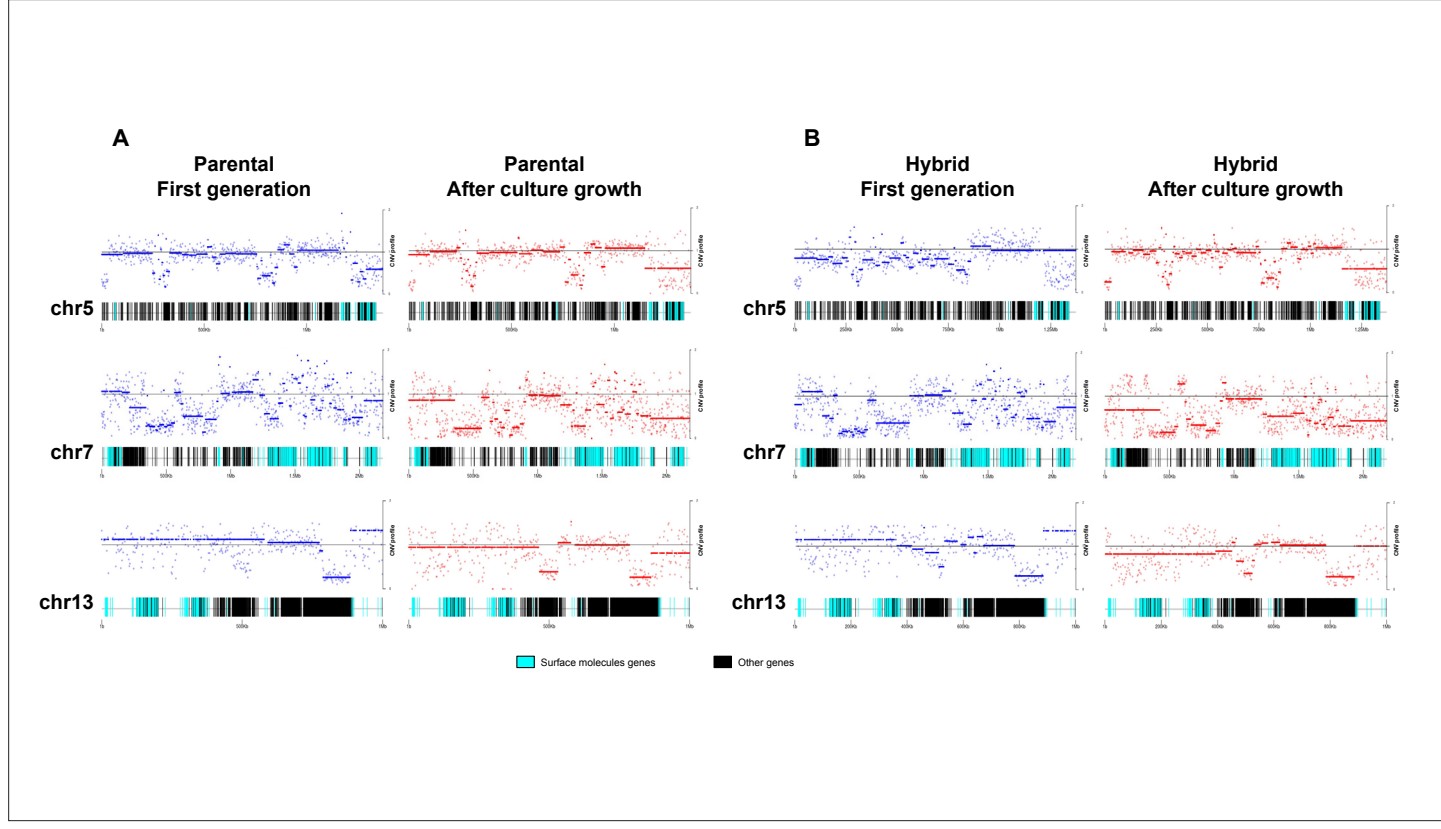

**Figure 6.** Chromosome gene composition and copy number variation (CNV) comparison indicate genome erosion patterns in surface molecule-coding regions after in vitro culture growth. (**A**) CNV comparison in parental strain. (**B**) CNV comparison in hybrid strain. Each dot in the CNV profile chart represents the normalized depth per kb, while lines represent the median ratio (*Cingolani et al., 2012*). Blue points and lines represent CNV profiles in the first generation, while red points and lines represent CNV profiles after 800 generations of in vitro culture growth. surface protein-coding genes are represented as cyan blue boxes, while other genes are represented as black boxes.

The online version of this article includes the following figure supplement(s) for figure 6:

**Figure supplement 1.** Counts of A. gene families and other membrane functional categories and B. other functional functional categories displaying copy number variation after culture growth.

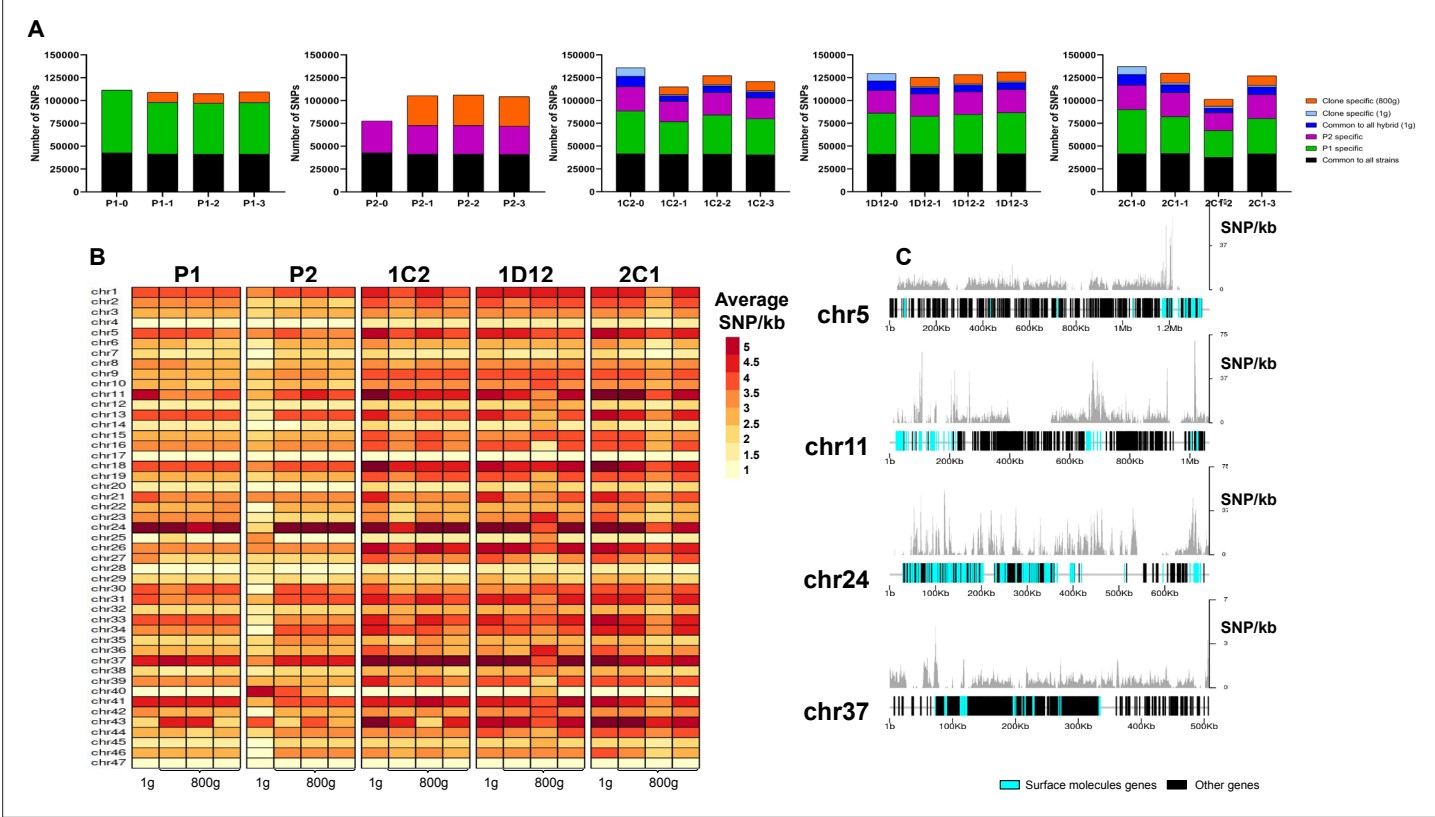

**Figure 7.** SNP density analysis throughout the genome indicates that surface molecule-coding regions displayed a higher number of variants than other regions of the genome for all strains. (**A**) Number of SNPs in all clones after long-term in vitro culture. In black are represented SNPs common to all hybrid strains, in green SNPs specific to P1 strain, in purple SNPs specific to the P2 strain, in dark blue SNPs common to all hybrids in the first generation, in light blue SNPs exclusive to each hybrid clone in the first generation, and in orange SNPs exclusive to each clone after culture growth. (**B**) Chromosome average SNP density per kilobase in all samples before (1 g) and after culture growth (800 g). (**C**) 1C2 chromosomes with higher SNP density plotted displaying gene composition and SNP density. Surface protein-coding genes are represented as cyan blue boxes, while other genes are represented as black boxes.

2.81; P2-3 2.66, Mann-Whitney, p=0.042) (*Figure 7A*). The P2 replicates have a reduction in copies of HUS1-like encoding genes, which are involved in DNA repair, after culture growth (*Supplementary file 12*). Our gene enrichment analysis showed that a number of these genes were located on chromosome 2 (*Supplementary file 12*), suggesting that the loss of trisomy in this chromosome could possibly have contributed to the higher number of mutations that was found in the P2 clones. New genomic variants were found in both evolved parental clones (*Figure 7A*). All evolved hybrid replicate clones, with exception of 1D12-3, showed a slight decrease in SNP density (1C2-0: 3.48; 1C2-1: 2.96; 1C2-2: 3.26; 1C2-3: 3.10; 1D12-0: 3.29; 1D12-1: 3.24; 1D12-2: 3.22; 1D12-3: 3.36; 2C1-0: 3.49; 2C1-1: 3.34; 2C1-2: 2.64; 2C1-3: 3.25, *Figure 7A*), likely caused by the loss of genetic material. Multiple new genomic variants were found in the in vitro evolved hybrid progeny, displaying patterns specific to each clonal isolate (*Figure 7A*). However, these variants were distinct from the novel mutations that emerged in the period between hybrid formation and the start of the microevolution experiment (t=0) as described above.

To better understand how these genomes evolved during culture growth, the SNP density was also evaluated in each chromosome separately. The SNP density pattern per chromosome remained similar for all samples in first generation (t=0) and after culture growth (t=800). Some chromosomes (e.g. chromosomes 5, 11, 24, and 37) display higher SNP density than the others across all samples (*Figure 7B*). P2 evolved clone replicates had an increase in SNP density in nearly all chromosomes (*Figure 7B*). The emergence of novel SNPs after culture growth resulted in most of the non-synonymous mutations. The ratio of non-synonymous per synonymous changes in novel SNPs was similar between both parental strains (P1-1: 2.26; P1-2: 2.22; P1-3: 2.38; P2-1: 2.25; P2-2: 2.22; P2-3: 2.26) and hybrid

clones (1C2-1: 2.20; 1C2-2: 2.48; 1C2-3: 2.26; 1D12-1: 2.41; 1D12-2: 2.5; 1D12-3: 2.40; 2C1-1: 2.21; 2C1-2: 2.18; 2C1-3: 2.30). Interestingly, the regions of the genome with higher genetic variation in all clone isolates contain surface protein-coding sequences (*Figure 7C*). To investigate the impact of new mutations in these regions, we compared the ratio of non-synonymous mutations in surface protein-coding genes to non-synonymous mutations in other genes. This ratio was higher for the mutations that appeared in each evolved parental clone in comparison to those present in the first generation (approx. 2.5-fold higher for P1 clones and 3.5-fold higher for P2 clones; *Supplementary file 13*). Regarding the hybrid strains, a similar ratio was observed in the SNPs that appeared in each evolved hybrid clone and those exclusive to each hybrid clone present in the first generation (*Supplementary file 13*). These results indicate that surface molecule-coding regions may mutate faster than the rest of the genome either by clonal or parasexual mechanisms, representing an important region of genomic diversification.

## Discussion

The results from whole genome sequencing analysis of this *T. cruzi* genetic cross have clearly demonstrated that, while the parental strains were diploid, all initial hybrid clones were essentially tetraploid. We showed that genomic variants exclusive to both parental strains were present in all hybrid clones, indicating that the tetraploid profile observed is obtained by the fusion of the two disomic parental genomes, although more complex pathways cannot be ruled out in the absence of direct observational evidence. In addition, we showed that genome erosion seems to be gradual, with sequential losses of chromosome copies rather than coordinated jumps between ploidy levels, though with some variation in erosion rates between different evolved hybrid lines. As previously proposed (*Lewis et al., 2011*; *Gaunt et al., 2003*), our results confirmed that hybridization in *T. cruzi* happened via a parasexual mechanism rather than a canonical meiotic process. This phenomenon resembles the parasexual cycle of the pathogenic fungus *Candida albicans*, which also involves diploid fusion followed by non-meiotic genome reduction back to aneuploids and diploids (reviewed by *Bennett, 2015*). In contrast, hybridization in *T. brucei* and possibly in *Leishmania* spp. has been associated with parental strains having undergone a meiosis-like process followed by a fusion of haploid cells (*Inbar et al., 2019*; *Peacock et al., 2014*; *Gibson et al., 2008*; *Rogers et al., 2014*; *Louradour et al., 2020*; *Van den Broeck et al., 2020*). In this process, mainly diploid hybrids are formed, but polyploid hybrids may also be produced by the fusion between parental cells that failed to undergo meiosis (*Inbar et al., 2019*; *Peacock et al., 2014*; *Gibson et al., 2008*; *Rogers et al., 2014*; *Louradour et al., 2020*; *Van den Broeck et al., 2020*). Despite the clear evidence of parasexual hybridization in in vitro *T. cruzi* hybrids, it is still not clear if a meiosis-like mechanism could also contribute to the generation of natural *T. cruzi* hybrids, as observed in other trypanosomatids. Further work will be required to establish whether there is a single common mechanism or several alternative modes of hybridization.

Our microevolution experiment has shown that *T. cruzi* genomes are highly responsive to the environmental conditions. The CNV analysis showed that while surface gene numbers were being eroded after culture growth, genes involved in cell metabolism, cell division, and ion transport were being expanded. This genome plasticity was not only observed at the level of gene CNV, but also at the chromosome level. The parental strains displayed distinct aneuploidies before and after culture, and despite the overall pattern of genome erosion, some chromosomes remained tetrasomic in the hybrid strains. Indeed, aneuploidies have been widely reported in natural *T. cruzi* populations, and although CCNV varies among and within DTUs, it seems constant within a given population (*Schwabl et al., 2019*; *Reis-Cunha et al., 2015*; *Reis-Cunha et al., 2018*). While in many multicellular organisms, aneuploidy is known to have severe consequences, in trypanosomatids it represents an important mechanism for rapidly overcoming environmental diversity (*Reis-Cunha et al., 2018*; *Prieto Barja et al., 2017*; *Bussotti et al., 2018*; *Grünebast and Clos, 2020*). Aneuploidy is reversed quickly once the benefits of an extra chromosome expire, as observed in P2 aneuploidies in chromosomes 2 and 21. In addition, trypanosomatids may control overexpression caused by additional gene copies, at the protein level (*Dumetz, 2017*). We found aneuploidy in chromosome 37 in all parental and hybrid clones after culture growth, suggesting a benefit to parasite fitness in culture. As observed in *Leishmania* spp., the fixation of genetic alterations is the result of positive selection processes that adapt parasite fitness to a given ecology or transmission cycle (*Bussotti et al., 2018*). We can speculate that in natural *T. cruzi* hybrids, distinct CCNV and genome erosion patterns would be observed due

to specific environmental selective pressure with the expansion of genes involved in host-parasite interactions.

Genome variant analyses showed that an increase in genetic sequence diversity was present after hybridization. Hybridization events are associated with a burst of novel mutations and a myriad of genomic rearrangements (e.g. transpositions, genome size changes, chromosomal rearrangements) in distinct eukaryotes (*Steensels et al., 2021*; *Samarasinghe et al., 2020*). Our results indicate that the process of *T. cruzi* hybrid formation included both new mutations and most likely recombination between parental chromosomes. Here, we were able to show an increase in the number of SNPs in the hybrid strains with the accumulation of amino acid substitutions. Interestingly, the numbers of new SNPs in culture showed a clear difference between the two parental strains after 800 generations, in which P2 showed a higher mutation rate. A previous study has reported that TcI strains have a mismatch repair machinery, which repairs base misincorporation and erroneous insertions and deletions during DNA recombination and replication, that is more efficient than in other DTUs (*Reis-Cunha et al., 2015*; *Augusto-Pinto et al., 2003*; *Machado et al., 2006*). As described in *Leishmania major*, HUS1 was shown to be required for genome stability under non-stressed conditions and its suppression had a genome-wide mutagenic effect (*Damasceno et al., 2018*). Since many of the genes coding for the HUS1-like proteins are encoded on chromosome 2, the loss of trisomy in this chromosome after culture growth could possibly be related to the increase in mutations observed in the P2 clones. In general, culture growth led to a reduction in genetic variants in nearly all evolved clones in comparison to the first generation. This reduction may be related to the loss of heterozygosity, but also to the loss of gene copies and genome erosion of surface molecules regions after culture growth. In addition, we could not identify any bias in the loss of a specific parental chromosome even in triploid hybrid clones, suggesting recombination between the parental chromosomes. As material from both parental strains has been lost during over 800 generations, it is likely that the hybrid clones would stabilize at diploidy and contain chromosomes from both parental strains, with a higher polymorphism than the parental strains. This is similar to strains from the known natural hybrid clades, TcV and TcVI. Strains from these clades have diploid genomes with high repeat content, and high levels of polymorphism in many regions, resulting from alleles from two divergent parental strains (from clades TcII and TcIII) (*Lewis et al., 2011*; *Messenger and Miles, 2015*; *Lewis et al., 2009*). From our analyses, it thus appears that the in vitro hybrid formation presented here is highly similar to the process that generated the natural hybrids.

Our microevolution experiment has shown that surface protein-coding genes represent the regions of the genome with higher genetic diversity and more rapid evolution. Accumulations of distinct genomic signatures and amino acid substitutions were observed in both parental and hybrid clones after relatively few generations. We have previously observed that there is a higher frequency of recombination in repeat-rich regions of the *T. cruzi* genome (*Talavera-López et al., 2021*), and it is likely that this is also the case for the hybrids.

We can speculate that this is a likely explanation for variability in repeat and gene copy numbers between hybrid and non-hybrid clades in response to the distinct environmental conditions. Since these parasites were cultivated solely in culture, it is still not clear if this mechanism is related to the lack of pressure caused by the progression within triatomine bugs or by the interaction with the host immune system, or if these genomes display an intrinsic mechanism of high diversification of these regions. Recent comparative genomics of natural *T. cruzi* strains have shown a similar pattern, suggesting that these regions indeed evolve more rapidly than other regions of the genome (*Wang et al., 2021*).

For *T. cruzi*, there is clear evidence for relatively recent inter-DTU hybridization between TcII and TcIII as the origin of TcV and TcVI, which are now widespread in domestic transmission cycles in southern South America (*Lewis et al., 2011*; *Messenger and Miles, 2015*; *Carrasco et al., 1996*). Other evidence for inter-DTU genetic exchange between TcI and TcIV comes from phylogenetic incongruence between nuclear and mitochondrial (kDNA maxicircle) sequences. However, the frequency of natural genetic exchange events between lineages/clades/DTUs appears to be too low to disrupt the overall stability of the main six clades, or seven clades if the TcBat lineage is included (*Lewis et al., 2011*; *Messenger and Miles, 2015*). The exception is the validity and position of the poorly sampled TcIV clade, which includes strains with TcI-like and/or TcIII-like features. The robustness of the other clades is likely promoted by ecological and/or geographical isolation, for example, association with

distinct vector and host species. There may also be a high fitness cost for tetraploidy in the wild that is not replicated in laboratory conditions.

In summary, we confirmed that in vitro hybrid formation in *T. cruzi* happened in a parasexual mechanism, representing an important mechanism of genetic diversification. Tetraploid hybrids showed patterns of progressive genome erosion shaped by the environment's selective pressure, which may explain distinct phenotypes in isolated natural hybrid populations. In addition, an increase in coding sequence diversity is observed in hybrids in comparison to the parental strains, related to both new mutations and most likely recombination events. Finally, our microevolution experiment has shown that repetitive gene families related to immune evasion evolved more rapidly than other regions of the genome, indicating an intrinsic mechanism of genetic variation of these regions.

## Materials and methods

### Parasites and microevolution experiment

For routine culture, *T. cruzi* epimastigotes were grown axenically in supplemented RPMI as previously described (*Carrasco et al., 1996*). Parasites were cloned by limiting dilution. The microevolution experiment comprised continuous in vitro cultures of each parasite line for 5 years. Three experimentally generated hybrid clones (1C2, 1D12, and 2C1) and their two parents (P1 and P2) were seeded in the primary culture at $2 \times 10^5$/ml and then allowed to grow for seven generations to stationary phase. The parent and hybrid lines used to initiate the microevolutions experiment (t=0) were estimated to be respectively 70 and 95 generations removed from the hybridization events. Parasites were passaged into fresh media every 2 weeks at a seeding density of $2 \times 10^5$/ml, equating typically to 1% of the stationary phase cultures.

### Flow cytometry

DNA content was determined as previously described (*Lewis et al., 2009*). In brief, mid-log phase epimastigotes were washed in PBS, then fixed overnight in ice-cold 70% methanol/30% PBS. Fixed cells were washed in PBS, adjusted to $1 \times 10^6$cells/ml, and incubated for 45 min with 10 µg/ml propidium iodide and 10 µg/ml RNAse A at 37°C. Fluorescence was detected using a FACSCalibur flow cytometer for a minimum of 10,000 events. FlowJo software (Tree Star Inc, Ashland, OR) was used to plot histograms and identify G1-0 and G2-M peaks. Mean G1-0 values were taken to infer relative DNA content. An internal control *T. cruzi* strain, Esm cl3, was included in each run. Relative DNA content values were calculated as ratios compared to the internal standard. For experimental hybrids, the ratios relative to each parent (P1 or P2) were also determined using the mean standard:parent ratios derived from 12 independent experiments.

### Microsatellite analysis

Genotyping was done for four microsatellite loci: MCLF10, 10,101(TA), 7,093(TC), and 10,101(TC) as previously described (*Lewis et al., 2009*). Briefly, microsatellites were PCR amplified from genome DNA samples using fluorescent-labelled primers targeting conserved flanking regions. Amplicon lengths were determined using a 48-capillary 3730 DNA analyzer (Applied Biosystems, UK) and analysed using Genemapper v3.5 software (Applied Biosystems, UK). The size of different PCR products (alleles), visualized as fluorescence peaks, were determined automatically by the software using a size standard to calibrate the calculations. All allele size calls made by the software were checked manually against a library of known TcI alleles (*Llewellyn et al., 2009*).

### Whole genome sequencing

Total genomic DNA was isolated directly from long-term parasite cultures using the Gentra Puregene Tissue Kit (Qiagen) according to the manufacturers' instructions. A total of 40 ng of genomic DNA was used as a template to prepare the sequencing libraries with the Rubicon Kit and the Illumina TruPlex kit. For the two parent strains, a 180 and 350 bp insert sizes paired end libraries were produced, plus an additional 8 kb insert size mate paired library. For the hybrid offspring, a single paired end library was produced with an insert size of 350 bp. All libraries were sequenced using the Illumina HiSeq 2500 platform. The library complexity of each parent was analysed using the 17mer distribution of the Illumina libraries using Jellyfish (*Llewellyn et al., 2009*).

## De novo genome assembly

The genomes of the parent strains were assembled using the 180 bp Illumina paired end library and the 8 kb Illumina mate pair library using the ALLPATHS-LG v52488 (https://software.broadinstitute.org/allpaths-lg/blog/) assembler with K=96, using the TcI Sylvio X10/1 reference strain (*Talavera-López et al., 2021*) for evaluations. The resulting contigs were scaffolded using both Illumina libraries and the BESST scaffolder (*Sahlin et al., 2014*). Later, gaps in the scaffolded assemblies were filled using the 180 bp with GapFiller (*Nadalin et al., 2012*) and chromosomes were numbered according to their sizes as for the human genome. Each genome was submitted for annotation, prior removal of repetitive elements using RepeatMasker (https://www.repeatmasker.org/), to the Companion annotation pipeline (*Steinbiss et al., 2016*).

## Data processing and read mapping

Paired end libraries were quality filtered using Nesoni Clip program (https://github.com/Victorian-Bioinformatics-Consortium/nesoni; *Harrison, 2020*) removing bases with a Phred quality score below 30 and reads shorter than 64 nucleotides; later, sequencing adaptors were removed. Mate paired libraries were processed as above, plus an additional step to reverse complement the filtered reads. Filtered reads, from all the sequencing libraries produced, were mapped against the TcI Sylvio X10/1 reference genome using Burrows-Wheeler Aligner (*Li and Durbin, 2009*). The mapping files were sorted, PCR and optical duplicates were removed, and read groups were added using Picard Tools v1.134 (https://broadinstitute.github.io/picard/; *Schreiber, 2022*). These mapping files were used for downstream analyses.

## Identification of genomic variation

SNPs were called using a mapping-based approach. Mapping files were sorted with Samtools v1.11, PCR-duplicates were marked with Picard v2.22.4 and SNPs were identified with Genome Analysis Toolkit (GATK – https://gatk.broadinstitute.org/hc/en-us) using the HaplotypeCaller algorithm with a minimum quality value of 30 and a minimum depth of coverage of 10. To avoid the influence of low mapping quality in repeated regions, a strict mapping quality filter was applied and any SNP in regions with mapping quality below 50 was removed using BCFtools (*Li et al., 2009*).

VCFtools package (*Li et al., 2009*) was used to infer SNP density per 1 kb (option --*SNPdensity*) and SNPs exclusive to each strain were identified using both VCFtools and BEDtools (*Li et al., 2009*; *Danecek et al., 2011*; *Quinlan and Hall, 2010*). The functional effect of these variants and the ratio of synonymous per non-synonymous mutations was predicted using SNPEff v4.4 (*Cingolani et al., 2012*).

## CCNV and ploidy estimation

CCNV and ploidy estimation were performed using a combination of RDC and AB methods as previously described (*Reis-Cunha and Bartholomeu, 2019*; *Reis-Cunha et al., 2015*). Briefly, the estimation of CCNV was based on the ratio between the mean coverage of predicted single-copy genes in a given chromosome and the mean coverage of all single-copy genes in the genome. This approach was based on the RDC of 2602 1:1 orthologs between the haplotypes of the reference SylvioX10/1 genome and the parental strains assemblies, identified using OrthoVenn2 (https://orthovenn2.bioinfotoolkits.net/home). For chromosomes lacking single-copy genes, CCNV was estimated based on the ratio between the mean RDC of each chromosome position and the mean coverage of all genome positions. If the ratio between the median chromosome coverage and the median genome coverage was approximately one, the chromosome had the same copy as the genome overall, while fluctuations in this ratio (lower than 0.75 or higher than 1.25) were putative aneuploidies.

Heterozygous SNPs with a mapping quality higher than 50 were selected for AB analyses to confirm aneuploidies and to estimate chromosome somy and whole genome ploidy. Whole genome ploidy was estimated by analysing the proportion of the alleles in heterozygous SNP positions of all single-copy genes. To confirm the RDC somy estimations, the proportion of each allele with heterozygous SNPs per chromosome position was plotted for 46 TcI chromosomes. All RDC and AB graphs were generated in R with ggplot2 (https://ggplot2.tidyverse.org), pheatmap (https://CRAN.R-project.org/package=pheatmap), and vcfR (*Knaus and Grünwald, 2017*) packages.

Genome erosion patterns were identified by CNV estimations using Control-FREEC package (*Boeva et al., 2012*). Gene Ontology Enrichment analyses in CNV regions were performed using

TriTryp tools (https://tritrypdb.org/tritrypdb/app/) and surface molecules genes were assigned from the available TcI Sylvio X10/1 annotation (*Talavera-López et al., 2021*) (available here).

## Acknowledgements

This study was financed in part by the Swedish Research Council, Dnr 20016–02951, and Coordenação de Aperfeiçoamento de Pessoal de Nível Superior – CAPES (Brazilian Government Agency) – Finance Code 001, GMM was supported by a scholarship provided by CAPES-PrInt and ECG was funded by grants from CNPq and CAPES-STINT and CAPES-PrInt (Brazilian Government Agencies). We are thankful to João Luís Reis Cunha for his guidance on ploidy estimation.

## Additional information

### Competing interests

The other authors declare that no competing interests exist.

### Funding

| Funder | Grant reference number | Author |
| --- | --- | --- |
| Swedish Research Council | Project Grant | Gabriel Machado Matos<br>Björn Andersson<br>Michael A Miles |
| CAPES | Student Scholarship | Björn Andersson<br>Edmundo C Grisard |

The funders had no role in study design, data collection and interpretation, or the decision to submit the work for publication.

### Author contributions

Gabriel Machado Matos, Formal analysis, Investigation, Methodology, Validation, Visualization, Writing – original draft, Writing – review and editing; Michael D Lewis, Conceptualization, Data curation, Formal analysis, Investigation, Methodology, Visualization, Writing – original draft, Writing – review and editing; Carlos Talavera-López, Data curation, Formal analysis, Investigation, Methodology, Software, Writing – original draft, Writing – review and editing; Matthew Yeo, Investigation, Methodology; Edmundo C Grisard, Formal analysis, Investigation, Supervision, Writing – original draft, Writing – review and editing; Louisa A Messenger, Data curation, Investigation; Michael A Miles, Conceptualization, Data curation, Formal analysis, Funding acquisition, Investigation, Project administration, Writing – original draft, Writing – review and editing; Björn Andersson, Conceptualization, Data curation, Formal analysis, Funding acquisition, Investigation, Project administration, Resources, Supervision, Validation, Writing – original draft, Writing – review and editing

### Author ORCIDs

Gabriel Machado Matos ● http://orcid.org/0000-0003-3744-2673
Björn Andersson ● http://orcid.org/0000-0002-4624-0259

### Decision letter and Author response

Decision letter https://doi.org/10.7554/eLife.75237.sa1
Author response https://doi.org/10.7554/eLife.75237.sa2

## Additional files

### Supplementary files

• Supplementary file 1. PCR-based multilocus microsatellite genotyping in parental and hybrid strains before (t=0) and after (t=800) the microevolution experiment.

• Supplementary file 2. Genome assembly statistics of parental strains.

• Supplementary file 3. Effects of SNPs in the first generation hybrids.

- Supplementary file 4. Number of SNPs in surface molecules genes (SM) and in other genes.

- Supplementary file 5. Number of non-synonymous SNPs in surface molecules genes (SM) and in other genes.

- Supplementary file 6. Gene Ontology analysis of expanded genes in parental strains after culture growth. Bgd count: Number of genes with this term in the genome; Result count: Number of genes with this term in our analysis; Pct of bgd: The percent of genes with this term in our analysis divided by the percent of genes with this term in the genome; Fold enrichment: Of the genes in the genome with this term, the percent that are present in your analysis.

- Supplementary file 7. Gene Ontology analysis of contracted genes in parental strains after culture growth. Bgd count: Number of genes with this term in the genome; Result count: Number of genes with this term in our analysis; Pct of bgd: The percent of genes with this term in our analysis divided by the percent of genes with this term in the genome; Fold enrichment: Of the genes in the genome with this term, the percent that are present in your analysis.

- Supplementary file 8. Gene Ontology analysis of expanded genes in hybrid strains after culture growth. Bgd count: Number of genes with this term in the genome; Result count: Number of genes with this term in our analysis; Pct of bgd: The percent of genes with this term in our analysis divided by the percent of genes with this term in the genome; Fold enrichment: Of the genes in the genome with this term, the percent that are present in your analysis.

- Supplementary file 9. Gene Ontology analysis of contracted genes in hybrid strains after culture growth. Bgd count: Number of genes with this term in the genome; Result count: Number of genes with this term in our analysis; Pct of bgd: The percent of genes with this term in our analysis divided by the percent of genes with this term in the genome; Fold enrichment: Of the genes in the genome with this term, the percent that are present in your analysis.

- Supplementary file 10. Gene Ontology analysis of genes encoded in chromosome 19. Bgd count: Number of genes with this term in the genome; Result count: Number of genes with this term in our analysis; Pct of bgd: The percent of genes with this term in our analysis divided by the percent of genes with this term in the genome; Fold enrichment: Of the genes in the genome with this term, the percent that are present in your analysis.

- Supplementary file 11. Gene Ontology analysis of genes encoded in chromosome 37. Bgd count: Number of genes with this term in the genome; Result count: Number of genes with this term in our analysis; Pct of bgd: The percent of genes with this term in our analysis divided by the percent of genes with this term in the genome; Fold enrichment: Of the genes in the genome with this term, the percent that are present in your analysis.

- Supplementary file 12. Gene Ontology analysis of contracted genes in Parental 2 clones after culture growth. Bgd count: Number of genes with this term in the genome; Result count: Number of genes with this term in our analysis; Pct of bgd: The percent of genes with this term in our analysis divided by the percent of genes with this term in the genome; Fold enrichment: Of the genes in the genome with this term, the percent that are present in your analysis.

- Supplementary file 13. Number of non-synonymous SNPs in surface molecules genes (SM) and in other genes after culture growth.

- Transparent reporting form

### Data availability

The data generated in this study have been submitted to the NCBI BioProject database (https://www.ncbi.nlm.nih.gov/bioproject/https://www.ncbi.nlm.nih.gov/bioproject/) under accession number PRJNA748998.

The following dataset was generated:

| Author(s) | Year | Dataset title | Dataset URL | Database and Identifier |
|---|---|---|---|---|
| Matos GM, Lewis MD, Talavera-López C, Yeo M, Grisard EC, Messenger LA, Miles MA, Andersson B | 2022 | Comparative genomic analyses of Trypanosoma cruzi experimental hybrids | https://www.ncbi.nlm.nih.gov/bioproject/PRJNA748998 | NCBI BioProject, PRJNA748998 |

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
