## [Editor Report]

The authors dissected the across-the-genome consequences of sexual recombination in *Trypanosoma cruzi*, a serious human pathogen. They had discovered hybrid formation in this species 20 years ago, here they went at length by culturing parental and hybrid clones for 5 years, demonstrating that tetraploid *T cruzi* hybrids undergo genome erosion.

---

## [Decision Letter]

**Decision letter after peer review:**

Thank you for submitting your article "Experimental microevolution of *Trypanosoma cruzi* reveals hybridization and clonal mechanisms driving rapid diversification of genome sequence and structure" for consideration by *eLife*. Your article has been reviewed by 3 peer reviewers, and the evaluation has been overseen by a Reviewing Editor and Jos Van der Meer as the Senior Editor. The following individuals involved in review of your submission have agreed to reveal their identity: Julius Lukes (Reviewer #1); Fernán Agüero (Reviewer #2).

Essential revisions:

1. The authors should revise their methods for SNP calling. What GATK variant caller was used? UnifiedGenotyper or HaplotypeCaller? The latter is recommended. It is also unclear which filtering criteria were used to reduce the likelihood of false-positive SNPs. Simply setting the mapping quality to a minimum of 50 is not sufficient. We recommend to have a look at the methods of Schwabl et al. 2019 (https://doi.org/10.1038/s41467-019-11771-z) where rigorous analyses were done to remove false-positive SNPs. In particular, it would be helpful to mask the reference genome based on the mappability of its own reads. This is also important for later sections in the manuscript where the authors claim increased diversity in repetitive surface molecule genes.

Editor: Also, it is not clear whether Indels were analysed at all, they appear once in the Methods and not in the rest of the manuscript.

2. Figure 1e, Table 1 and corresponding main text – We are puzzled regarding the number of SNPs that are unique to each hybrid clone and that are common to all hybrids.

First, it seems to us that the authors have done SNP calling in all strains independently, which may result in genotypes being missed because of low local coverages. Joint genotyping would mitigate such biases. We would recommend to do a joint genotyping across all parental and hybrid strains, and then only retain those SNP positions for which there is no missing data.

Second, in the main text on page 7, it is argued that these SNPs would have occurred in culture during the approximately 50 generations of growth BEFORE hybrid formation. But in the last sentence of that paragraph it is stated that these SNPs appeared after the hybridisation event. How can the authors tell whether these SNPs occurred during culture before hybridization, or whether they happened just after hybridization? The setup of the experiment should be more clearly explained: how many generations happened before hybridization, how many after, when exactly was sequencing done during this process? This is also important for later sections that claim that hybridization resulted in a burst of novel mutations.

It seems like the authors are discovering the impact of tandem repeat regions (copy number variations) on SNP discovery. Most of page 10 and Figure 3 merely describes the fact that you will have more false-positive SNPs within the repeat regions. I'm not sure why this is of interest here. Such SNP's should not have been included in the first place.

3. Why was the mapping quality in surface molecules inferior to the rest? That begs for explanation. "Surface molecule genes", frequently used here, sounds somewhat weird to this reviewer. Couldn't it be rather "Surface protein-coding genes" or the like?

4. I understand the implications of trying to map short reads to repetitive regions. For example, PAINTS (see https://www.ncbi.nlm.nih.gov/pmc/articles/PMC7912377/) removes noisy genomic regions (large gene families) before attempting to do ploidy estimations. This seems intuitive, and this is also explained in this Ms, in light of this maybe authors can save some space by omitting text and figures that are used to discuss and show these difficulties? Maybe there is not much to gain by showing that estimating ploidies in chromosomes with low numbers of housekeeping genes is difficult. Just state that and put the data into Supplementary file?

5. Based on the results presented, I don't agree that there is gradual erosion overall across all clones.

1D12 hybrid seems to have gone through an initial loss of DNA content but then its DNA content seems to have remained stable (Figure 1D). In addition, it's genome-wide somy profile remained stable at tetrasomy (Figure 4C). Also, there also seems to be an increase in DNA content after 600 generations. Finally, microsatellite alleles are only consistently lost in hybrid 2C1, at three different loci. No loss was observed in hybrid 1D12 and only one allele was lost in hybrid 1C2. This also raises the question what has caused the loss of DNA content in hybrid 1D12. Based on microsats and somy, there is no clear evidence for genomic erosion. So, what part of the genome is lost in the beginning of the experiment? The authors could do a comparative genomic analyses of the parental genomes and the hybrid genomes, to figure out what portions of the genomes are commonly/uniquely lost in the hybrids.

6. One of the conclusions from the study of accumulation of mutations, is that surface molecules may mutate faster. The reasoning for this if I read and understood correctly, is that there were many "mutations that appeared in each parental clone after culture growth". Do authors mean to say that these mutations appeared after the genetic cross experiment in Gaunt et al. 2003? Can the authors clarify how these parental clones were kept and passaged since then? How many passages did they undergo before being sequenced? The same clarification is warranted for hybrids. When the authors state that "while the parental strains were diploid, all initial hybrid clones were essentially tetraploid" (Discussion, page 15), the readership would like to know if these essential tetraploid states were observed just after the hybridization (genetic exchange) event (e.g. clones were kept in liquid nitrogen since then and only thawed for brief culture before sequencing) or if these states had some passages in culture after the genetic exchange event (how many?).

7. Data and code availability: The mentioned accession number was not found in the BioProject database. Please make code available via GitHub or other repository.

---

## [Author Response]

Essential revisions:1. The authors should revise their methods for SNP calling. What GATK variant caller was used? UnifiedGenotyper or HaplotypeCaller? The latter is recommended. It is also unclear which filtering criteria were used to reduce the likelihood of false-positive SNPs. Simply setting the mapping quality to a minimum of 50 is not sufficient. We recommend to have a look at the methods of Schwabl et al. 2019 (https://doi.org/10.1038/s41467-019-11771-z) where rigorous analyses were done to remove false-positive SNPs. In particular, it would be helpful to mask the reference genome based on the mappability of its own reads. This is also important for later sections in the manuscript where the authors claim increased diversity in repetitive surface molecule genes.Editor: Also, it is not clear whether Indels were analysed at all, they appear once in the Methods and not in the rest of the manuscript.

We thank the reviewers for highlighting these issues. These are important points for this complex genome and we agree that they should be clear. HaplotypeCaller was used, and more information on the SNP calling was added in the revised manuscript. We are familiar with the suggested methods, since the corresponding author co-authored the Schwabl et al. (2019) paper. We have now used the masking used in that paper to compare the two strategies. Only a small number of SNPs are lost when the masking is added, and the distribution is unchanged. Thus, the conclusions are not changed. In addition, we found that our strategy removed a larger proportion of the genome from the analysis than the masking strategy. As the strains used in the hybrid strategy differ significantly from the reference in many repetitive regions, it is likely that the strategy using only the parent/hybrid reads gives more accurate results. The strict mapping, combined with visualization of the mapping quality and comparing with the repeat content, has enabled us to more reliably remove unreliable SNPs in these extremely repetitive regions. As we produced the reference genome (Talavera-Lopez et al., 2021), we have performed extensive mapping of the reference reads to the sequence to identify repetitive regions. Indels were not found to be informative.

2. Figure 1e, Table 1 and corresponding main text – We are puzzled regarding the number of SNPs that are unique to each hybrid clone and that are common to all hybrids.First, it seems to us that the authors have done SNP calling in all strains independently, which may result in genotypes being missed because of low local coverages. Joint genotyping would mitigate such biases. We would recommend to do a joint genotyping across all parental and hybrid strains, and then only retain those SNP positions for which there is no missing data.

As explained below, these results are not caused by the SNP calling. As the coverage is high, joint genotyping would add significantly to the analysis. Moreover, the independent clone/parental strain analysis would not cause biased SNP detection, since a detailed analysis has compared the same regions of each sequenced genome, being consistent among the raw reads despite the sequencing coverage.

Second, in the main text on page 7, it is argued that these SNPs would have occurred in culture during the approximately 50 generations of growth BEFORE hybrid formation. But in the last sentence of that paragraph it is stated that these SNPs appeared after the hybridisation event. How can the authors tell whether these SNPs occurred during culture before hybridization, or whether they happened just after hybridization? The setup of the experiment should be more clearly explained: how many generations happened before hybridization, how many after, when exactly was sequencing done during this process? This is also important for later sections that claim that hybridization resulted in a burst of novel mutations.

We agree that more detail on sample origins and hybrids formation is required, though exact passage numbers from the original crossing experiment (performed in 1998) were not recorded, so these are best estimates from a revised assessment of all the available information (not included in the main text).

P1 and P2 clones were transfected with the drug resistance genes on episomal constructs, drug selected and polyclonal populations were grown out (~15 generations). Cloning generated the P1-hyg and P2-neo lines (~25 generations). Stationary phase cultures containing metacyclic trypomastigotes were used for the mixed Vero infections (crosses) as described in Gaunt et al., 2003 (~10 generations to point of mixing). The P1 and P2 samples that were sequenced in this study, for t=0, were grown from the pre-transfection parasites, adding another ~20 generations of culture. Thus, the t=0 parent sequences are removed by ~70 generations from the experimental cross.

Parasites were recovered from mixed Vero cell infections between 7 and 28 days post-inoculation (~15 generations as amastigotes), then selected (on both drugs) and a polyclonal population was grown out (~15 generations). Cloning generated the original 6 hybrid lines (~25 generations), which were cryopreserved. When work resumed on analysing the hybrids’ DNA contents (see Lewis et al., 2009) the t=0 DNA extraction occurred after ~40 generations of epimastigote culture. Thus, the t=0 hybrid sequences are removed by ~95 generations from the experimental cross. So, at the beginning of the microevolution experiment presented in this study, the parents and hybrids were already separated by approximately 165 generations. Figure 1 and the methods section has now been modified to reflect these details.

In hindsight, it would have been better to have less divergence between the time of DNA sampling and the original cross experiment, but such a long-term microevolution experiment was not planned in advance and the idea developed over time. However, it would never have been possible to avoid some distance between the parent samples, the hybrid clone samples and the actual hybrisation event because of the culture bottlenecks inherent to the process of cell infections, drug selection and cloning, as well as growing out sufficient numbers of parasites to yield enough DNA for the sequencing technologies available at the time.

In light of the above-mentioned aspects, we consider that SNPs that are common to all hybrids were all almost certainly generated from before hybridisation, falling into two main categories: SNPs that were detected in the three t=0 hybrid samples but not detected in the parent samples, and SNPs occuring in the parent parasite lines during culture before the cross was set up. As above, DNA for this study was derived from parent samples approximately 70 generations separating them from the cross.

We might rule out less likely origins for SNPs common to all hybrid such as homoplasies, that would be extremely rare across three independent hybrid lines, or those occuring during growth of a hypothetical common progenitor hybrid before the cloning procedure was done. This latter seems unlikely because hybrid ID12 inherited P2 kDNA maxicircles while 1C2 and 2C1 inherited P1 kDNA maxicircles. Thus, a minimum of two independent hybridisation events is more parsimonious than a mixed maxicircle common progenitor. It is noteworthy to mention that maxicircle inheritance is uniparental in *T. brucei* and *Lesishmania*.

The hybrid clone-specific SNPs are assumed to all be downstream of the hybridisation events, though it is conceivable that a very small fraction might have been undetectable in all samples but one for technical reasons. These are the SNPs being referred to at the end of this section where we state “this ratio was approximately 3.3-fold higher in the SNPs that appeared after the hybridization event”

We thus changed on page 7 “50 generations of growth before hybrid formation” to “70 generations of *parent parasite* growth before *the cross was set up*“ and added a sentence to explain about the kDNA. Figure 1 was changed to show the time separating the DNA samples from the hybridisation event. We also clarify at the end of section that SNPs that appeared after hybridisation equates to hybrid clone-specific SNPs.

It seems like the authors are discovering the impact of tandem repeat regions (copy number variations) on SNP discovery. Most of page 10 and Figure 3 merely describes the fact that you will have more false-positive SNPs within the repeat regions. I'm not sure why this is of interest here. Such SNP's should not have been included in the first place.

As the reviewers are already aware, the *T. cruzi* genome is unusual, even for kinetoplastid parasites. It is effectively divided into two parts, core and repetitive, and the two behave very differently in most respects, both regarding sequence analysis and biology. We have therefore chosen to include this information in order to illustrate the features of the genome for readers are less familiar with this organism and its genome, specially concerning the sequence variability within the multicopy gene families of a clonal organism that would allow detection of distinct SNPs among distinct members of a given gene family. We feel that this will be helpful, as *T. cruzi* is radically different from other trypanosomatids and it is suited for the readership of this more broad journal. We have rephrased the text to reflect to clarify this.

3. Why was the mapping quality in surface molecules inferior to the rest? That begs for explanation. "Surface molecule genes", frequently used here, sounds somewhat weird to this reviewer. Couldn't it be rather "Surface protein-coding genes" or the like?

The difference in mapping quality thus reflects the extreme nature of the sequence composition of the surface-molecule coding gene-containing regions of the genome. Even though rigorous mapping parameters were used, results such as these are expected due to the large and variable number of gene copies and other repeats that cannot reliably be used for mapping. We have changed the terminology as suggested.

4. I understand the implications of trying to map short reads to repetitive regions. For example, PAINTS (see https://www.ncbi.nlm.nih.gov/pmc/articles/PMC7912377/) removes noisy genomic regions (large gene families) before attempting to do ploidy estimations. This seems intuitive, and this is also explained in this Ms, in light of this maybe authors can save some space by omitting text and figures that are used to discuss and show these difficulties? Maybe there is not much to gain by showing that estimating ploidies in chromosomes with low numbers of housekeeping genes is difficult. Just state that and put the data into Supplementary file?

As mentioned above, we included and opted for maintaining such information in order to illustrate the unique features of the *T. cruzi* genome for readers are less familiar with this organism and its genome. We consider that this will be useful and helpful, as comprehension of distinct *T. cruzi* DTU genome structures have been impaired by a number of variables such as the number of multicopy gene families and their contents.

5. Based on the results presented, I don't agree that there is gradual erosion overall across all clones.1D12 hybrid seems to have gone through an initial loss of DNA content but then its DNA content seems to have remained stable (Figure 1D). In addition, it's genome-wide somy profile remained stable at tetrasomy (Figure 4C). Also, there also seems to be an increase in DNA content after 600 generations. Finally, microsatellite alleles are only consistently lost in hybrid 2C1, at three different loci. No loss was observed in hybrid 1D12 and only one allele was lost in hybrid 1C2. This also raises the question what has caused the loss of DNA content in hybrid 1D12. Based on microsats and somy, there is no clear evidence for genomic erosion. So, what part of the genome is lost in the beginning of the experiment? The authors could do a comparative genomic analyses of the parental genomes and the hybrid genomes, to figure out what portions of the genomes are commonly/uniquely lost in the hybrids.

We agree that the three hybrid lines have distinct genome erosion patterns not fully captured by our use of the term “gradual”. Revision of figure 1C to account for kDNA and infer 3n1k and 4n1k levels has allowed us to better visualise these trends. This shows there was in fact little DNA loss between the hybridisation (diploid fusion) events and t=0 for all three lines. While 2C1 shows the most consistent rate of genome erosion, with all three 800 generation evolved clones approximating a 3C genome size, the 1C2 shows more of a “stop-start” pattern, where most of the erosion occurs between 0-200 and 400-600 generations, reaching a 3C genome size, but two of the 800 generation clones then gain ~10%. The 1D12 clone, as the reviewer correctly observes, shows an initial 9% loss between t=0 and t=200 but was then quite stable between 3C and 4C.

In terms of the microsatellite data, we only typed four loci so there is very limited value in using them to interpret the total DNA erosion patterns. The purpose of such typing was just to help guide decisions on the analytical approach, i.e., whether to continue culturing for more years, whether to pursue a whole genome sequencing approach and for which samples. We reasoned that once we saw any evidence of microsattellite allele dropout then sequencing was likely to be justified.

Comparative genomic analyses of the parental genomes and the hybrid genomes is described on section “Patterns of genome erosion after in vitro microevolution” (Pg. 13) and we have now expanded the text of the Results section relating to Figure 1C and 1D to cover these aspects of the data and provided more nuance. Also, Gene Ontology analysis of contracted genes in hybrid strains after culture growth is presented in Supplementary Table 9.

6. One of the conclusions from the study of accumulation of mutations, is that surface molecules may mutate faster. The reasoning for this if I read and understood correctly, is that there were many "mutations that appeared in each parental clone after culture growth". Do authors mean to say that these mutations appeared after the genetic cross experiment in Gaunt et al. 2003? Can the authors clarify how these parental clones were kept and passaged since then? How many passages did they undergo before being sequenced? The same clarification is warranted for hybrids. When the authors state that "while the parental strains were diploid, all initial hybrid clones were essentially tetraploid" (Discussion, page 15), the readership would like to know if these essential tetraploid states were observed just after the hybridization (genetic exchange) event (e.g. clones were kept in liquid nitrogen since then and only thawed for brief culture before sequencing) or if these states had some passages in culture after the genetic exchange event (how many?).

Please refer to the answer to point #2 above about the origin and timing of the DNA samples. We have added text to clarify this issue.

7. Data and code availability: The mentioned accession number was not found in the BioProject database. Please make code available via GitHub or other repository.

The sequence data was submitted to the NCBI BioProject database under accession number PRJNA748998 and will be publicly available upon publication as now stated on the MS. URL: https://dataview.ncbi.nlm.nih.gov/object/PRJNA748998?reviewer=ejap1p42k4u6baufrhtpgls5e2.

The scripts, mostly used for plotting the figures in “R” are relatively standard and we do not think they warrant inclusion.